# Bacterial Diversity Analysis and Evaluation Proteins Hydrolysis during the Acid Whey and Fish Waste Fermentation

**DOI:** 10.3390/microorganisms9010100

**Published:** 2021-01-04

**Authors:** Alba C. Mayta-Apaza, Israel García-Cano, Konrad Dabrowski, Rafael Jiménez-Flores

**Affiliations:** 1Department of Food Science and Technology, Parker Food Science and Technology Building, The Ohio State University, Columbus, OH 43210, USA; maytaapaza.1@buckeyemail.osu.edu (A.C.M.-A.); garciacano.1@osu.edu (I.G.-C.); 2School of Environment and Natural Resources, The Ohio State University, 473D Kottman Hall, 2021 Coffey Rd., Columbus, OH 43210, USA; dabrowski.1@osu.edu

**Keywords:** acid whey, fish waste, *Lactobacillus rhamnosus*, proteolytic activity, microbial diversity

## Abstract

The disposal of acid whey (Aw), a by-product from fermented products, is a problem for the dairy industry. The fishery industry faces a similar dilemma, disposing of nearly 50% of fish processed for human consumption. Economically feasible and science-based alternatives are needed to overcome this problem. One possible solution is to add value to the remaining nutrients from these by-products. This study focuses on the breakdown of nutrients in controlled fermentations of Aw, fish waste (F), molasses (M), and a lactic acid bacteria (LAB) strain (Lr). The aim was to assess the dynamic variations in microbial diversity and the biochemical changes that occur during fermentation. Four treatments were compared (AwF, AwFM, AwFLr, and AwFMLr), and the fermentation lasted 14 days at 22.5 °C. Samples were taken every other day. Colorimetric tests for peptide concentrations, pH, and microbial ecology by 16S-v4 rRNA amplicon using Illumina MiSeq were conducted. The results of the microbial ecology showed elevated levels of alpha and beta diversity in the samples at day zero. By day 2 of fermentation, pH dropped, and the availability of a different set of nutrients was reflected in the microbial diversity. The fermentation started to stabilize and was driven by the Firmicutes phylum, which dominated the microbial community by day 14. Moreover, there was a significant increase (3.6 times) in peptides when comparing day 0 with day 14, making this treatment practical and feasible for protein hydrolysis. This study valorizes two nutrient-dense by-products and provides an alternative to the current handling of these materials.

## 1. Introduction

A massive shift in consumer trends has moved the production of dairy-based foods toward fermented products, including fresh and soft cheeses, sour cream, Greek-style yogurts, and caseinates. The increased production of these fermented products caused the increase of a by-product stream known as acid whey [1,2]. Acid whey is produced either by acidification from a starter culture composed of lactic acid bacteria (LAB) or by the addition of organic acids [3,4,5]. Acid whey is the green-yellowish serum phase of milk, which results from the precipitation and removal of caseins and contains the remaining lactose, whey proteins, fat, and minerals [6,7]. The high volume of whey and the efficient productivity of dairy plants make managing this by-product an urgent challenge [4]. Acid whey is currently directly disposed, mixed with manure and used as fertilizer, or given as a feed supplement for pigs or cattle; attempts to use it for biogas production have also been tested [1,2,3,4]. Due to the relatively high organic load in acid whey, the handling has become an economic and environmental burden; therefore, new alternative physicochemical technologies are needed to pre-treat or valorize it [7].

The fisheries industry also generates several waste streams that are nutritionally dense and have great potential for refinery by-products. In this sense, another by-product in need of alternative treatment is fish waste from fillet processing. Fish waste is highly perishable, and its natural degradation is associated with the growth of pathogenic bacteria. It represents more than 50% of the total weight from the production of fillets [8,9]. Most of this waste consists of bone frames, viscera, skin, scales, and in some cases, the whole fish when it is not suitable for further processing [10]. There have been some attempts to valorize this waste by utilizing it for animal feed [11,12,13]. However, if not properly processed, the by-products from the fishery industry are fast to degrade by endogenous enzymatic reactions, microbial spoilage, and oxidation—hence, the urgent need for this industry to develop a cost-effective alternative treatment [14,15]. In this work, the value of utilizing this by-product comes from the protein quality and quantity, which can yield protein hydrolysates in addition to providing storage stability and improvement in its nutritional bioavailability. The hydrolysates sought in this work are the result of proteins broken down by bacterial fermentation. These hydrolysates have proteases of known activity, and the fermentation process reduces them into peptides or free amino acids of value in feed production for aquaculture and, perhaps, other commercially essential crops [10,16].

LAB evolved to be fierce competitors in several microbial communities because the bacteria possess a dynamic and active metabolic system. Some LAB produce a variety of metabolites associated with enzymatic reactions, such as proteases, lipases, peptidoglycan hydrolases, and bacteriocins, among others [17]. The LAB proteolytic system allows for hydrolysis of large proteins; however, it is strain-specific and also depends on the conditions of the matrix in terms of substrate availability, pH, temperature, and the stage of the bacterial growth curve [14,18]. LAB opens up opportunities in the transformation of by-product streams from the agro-industrial sector, which are typically rich sources of nutrients.

The implementation of this type of biological technology provides alternative ways for several of these by-products to be treated [19,20]. For example, LAB can be used to break down collagen structures from the carcasses of the meat industry, and specialized compounds from lignocellulosic biomass fermentation can produce metabolites that can be implemented in the elaboration of detergents [21,22,23]. As described, there is an urgency to find solutions to the putrefaction of these waste materials and be able to leverage this dense nutritional availability. In this work, we focus on fermentation because this process occurs under controlled conditions and typically uses a specific microorganism to induce enhanced proteolytic degradation. This degradation results in the production of reduced molecular weight peptides, which are desirable in the formulation of feed diets [8]. The present work seeks to add value to the remaining nutrients from by-products of the dairy and fishery industry. We utilize these by-products as a source of recoverable nutrients in the form of protein hydrolysates throughout a fermentation process, thus allowing sustainable management of these resources. This study focused on the breakdown of complex proteins in a semi-controlled fermentation system and assessed the dynamic changes of the system’s biochemical components and microbial community shifts using microbial ecology tools.

## 2. Materials and Methods

### 2.1. Biological Materials

The acid whey was obtained from a local dairy processing plant (Superior Dairy, Canton, OH, USA). No additional processing was done before use in the fermentation system. Once collected, the material was stored at −20 °C. Silver carp (*Hypophthalmichthys molitrix*), mean weight 1.4 ± 0.5 kg, was originally obtained from the Illinois River in June 2018, transported alive to Columbus, OH, and then frozen and stored at −18 °C. Then, the whole fish was homogenized in a meat grinder after thawing and used for fermentation. Cane molasses (Groeb Farms Inc. Onsted, MI, USA) was used as a supplemental source of carbohydrate. We used a single strain starter, *Lactobacillus rhamnosus* OSU-PECh-69 [17]. Before inoculation, it was reactivated using 10 μL of preserved cells in 10 mL of De Man, Rogosa and Sharpe (MRS) broth (BD Difco; Franklin Lakes, NJ, USA) and incubated at 37 °C overnight.

### 2.2. Selection of Proteolytic Strain

For this work, we selected one strain from a total of 137 LAB strains from the OSU-PECh (Ohio State University-Parker Endowed Chair) bacteria collection. Each strain was evaluated by measuring its proteolytic activity in acidic conditions (the method is described in Section 2.4). The cell and the cell-free extract were evaluated to find the strain with the highest activity. *Lactobacillus rhamnosus* OSU-PECh-69 showed the highest activity (data not shown).

### 2.3. Fermentation

Acid whey, fish, and molasses were thoroughly mixed until a homogeneous mixture was achieved. Then the mixture was divided into separate containers. *L. rhamnosus* (OSU-PECh-69) was used as inoculum and added in the respective treatments. A total of four treatments were assessed, and the fermentation ratios are described in Table 1. The fermentation was carried out using a laboratory-scale fermenter (1 L glass carboy) and an S-shaped airlock stopper. All mixes were ‘incubated’ at room temperature with continuous stirring under aerobic conditions. Samples of 10 g of the wet weight were collected every other day from day 0 to day 14. Collected samples were stored at −80 °C until further assessment was performed.

### 2.4. Proteolytic Activity

Overnight culture (*L. rhamnosus* OSU-PECh-69) was centrifuged at 4122× *g* for 15 min at 4 °C (Sorvall Legend XF; Thermo Scientific, Waltham, MA, USA) to separate cells from the supernatant and then adjusted to 3.6 × 10^6^ CFU/mL. Proteolytic activity was determined as described by Anson [24] with some modifications. Briefly, an acidic buffer containing acetic acid, boric acid, and phosphoric acid at 0.025 M of final concentration at pH 5 was mixed with 0.5% of bovine blood hemoglobin (Sigma-Aldrich, St. Louis, MO, USA). The solution was sterilized using a 0.10 μm filter (Millipore, Burlington, MA, USA) and stored at −20 °C until use. Each reaction contained 300 μL of the hemoglobin solution and 100 μL of sample. The mix was incubated at 37 °C for 1 h. The reaction was stopped by adding 100 μL of trichloroacetic acid (50% w/v) and cooled at 4 °C for 10 min. The mix was centrifuged at 16,000× *g* for 10 min at room temperature, and 200 μL of supernatant was transferred in a 96-well plate (Flat bottom; Corning, Corning, NY, USA). Finally, the samples were read at 280 nm in a microplate reader spectrophotometer (Multiskan GO; Thermo Scientific, Walthman, MA, USA). One unit of proteolytic activity was defined as a change of 0.01 absorbances per min (U/min). The specific activity was correlated with the protein concentration (U/min × mg protein).

### 2.5. pH Measurement and Bacterial Counts

The pH of the fermentation was monitored using a SevenCompact™ pH/Ion Benchtop Meter (Mettler Toledo, Columbus, OH, USA) every other day. In addition, the fermentation was monitored by measuring LAB and total coliforms. For LAB and total coliforms, MRS (BD Difco, Franklin Lakes, NJ, USA) agar with bromocresol green (Sigma-Aldrich, St. Louis, MO, USA) as a pH indicator and EMB (Eosin-Methylene Blue; BD Difco, Franklin Lakes, NJ, USA) were used, respectively. Plates were incubated 24–48 h at 37 °C under aerobic conditions, after which time colonies were counted and expressed as colony-forming units per gram of sample (CFU/g).

### 2.6. Protein Analyses

The collected samples were separated by centrifugation (Centrifuge 5804R, Eppendorf, Hamburg, Germany) at 16,000× *g* for 15 min at room temperature. Carefully, the supernatant was collected and used for protein and peptides quantification.

#### 2.6.1. Protein Concentration Assay

The protein concentration was performed in the soluble fraction following the Bradford protein assay kit instructions (Bio-Rad; Hercules, CA, USA). The protein calculation was based on a standard curve using bovine serum albumin and expressed as a microgram per milliliter (μg/mL).

#### 2.6.2. Free Amino Acids and Peptide Analysis

The amino acid/peptide analyses were performed following the Cadmium-ninhydrin method described by Doi [25] with slight changes. The Cadmium-ninhydrin reagent was first prepared as follows: 0.8 g of ninhydrin, 80 mL of ethanol, 10 mL of glacial acetic acid, 1 g of cadmium chloride (Fisher Scientific; Hampton, NH, USA), and 1 mL of distilled water. The reaction was performed using 50 μL of sample plus 100 μL of Tris-HCl pH 8 buffer and 150 μL the Cd-ninhydrin reagent. The sample, buffer, and reagent were mixed carefully mixed and incubated at 84 °C for 5 min. Then, it was transferred to ice for 5 min and centrifuged at 10,000× *g* for 5 min. Finally, 200 μL of the reaction were transferred into a 96-well plate and read at 507 nm using a microplate reader spectrophotometer. The peptide calculation was done using a standard curve with bovine serum albumin and protease expressed as milligram per milliliter and normalized to percentage.

### 2.7. Microbial Community Analyses

#### 2.7.1. DNA Extraction and Quality Assessment

Microbial genomic DNA was extracted and purified directly from the fermentation product using DNeasy^®^ PowerSoil^®^ Kit (Qiagen, Hilden, Germany) following the manufacturer’s recommendations with a preliminary rinse with sterile phosphate-buffered saline (PBS, Gibco, Walthman, MA, USA). The mixed reaction was centrifuged at 10,000× *g* for 10 min. The pellet obtained was used for future work. Samples representing time-points (0, 2, 8, and 14 days) were selected for sequencing. Genomic DNA yield and purity were determined using a micro-drop spectrophotometer (Multiscan Go, Thermo Fisher, Walthman, MA, USA) 260/280 absorbance ratio and a 0.8% agarose (Sigma-Aldrich, St. Luis, MI, USA) gel, respectively.

#### 2.7.2. DNA Library Preparation and Sequencing

Further genomic DNA quality evaluation regarding integrity/fragmentation was evaluated by Diversigen Inc. (Huston, TX, USA), where the samples were sent for sequencing. After the required quality of samples was assessed, the genomic DNA was prepared into libraries for sequencing by Nextera DNA Flex Library Preparation Kit (Illumina, Catalog No. 20018705) using Nextera Index Kit (Illumina, Catalog No. FC-121-1012). Quality appraisal of the library quantification and size estimation was determined using the fragment analyzer electrophoresis system (Advanced Analytical Technologies, Inc.). Then, sequencing was performed on an Illumina MiSeq (2 × 250 bp).

#### 2.7.3. 16S-v4 Annotation

The pipeline used to analyze the 16S data integrates alignment-based and phylogenetic approaches to maximize the data results. The raw data were first demultiplexed into read 1 and read 2 based on their unique molecular barcodes built for library preparation. Subsequently, each was denoised and merged using DADA2 software [26], and then subject to chimera removal using VSEARCH [27]. Afterward, 16S rRNA gene sequences were clustered into Operational Taxonomic Units (OUTs) with a 97% cutoff value of similarity. The taxonomic identities were allocated to every OTU, utilizing a sci-kit-learn classifier and an optimized, variable region-specific version of the SILVA database [28]. Custom scripts constructed the rarefied-OUT table from the output files generated in the previous steps, which then were used to evaluate phylogenetic trends and analyses of alpha-diversity and beta-diversity [29]. Downstream statistical analysis and construction of visualization output were executed using a mix of public and proprietary packages in R.

### 2.8. Statistical Analyses

The collected data was fit to a completely randomized design with four treatments and three replicates. Analysis of variance (ANOVA) and mean differences were evaluated using the Tukey HSD test (*p* < 0.05). Statistical analysis was performed using JMP^®^ Pro 14 statistical software. The microbial community data were analyzed using QIIME 2 and public and proprietary packages in R.

### 2.9. Accession Number

The LAB used in this work was isolated from provolone cheese and identified as *Lactobacillus rhamnosus* OSU-PECh-69 (Gen Bank accession number: MT337424).

## 3. Results

### 3.1. pH and Microbiology Community

Acid whey from cottage cheese had a pH of 4.7 ± 0.18 and fresh minced and homogenized fish had a pH of 6.2 ± 0.36. The initial pH of the mix of acid whey and fish waste was 5.8 ± 0.29. As shown in Figure 1, in general, all the treatments showed a significant (*p* < 0.05) drop in pH by day two when compared to its respective initial pH. The treatment AwFMLr (Figure 1D) reached the lowest pH among all treatments (4.53 ± 0.001) and remained constant from the second day until the last day of fermentation. Treatment AwFM and AwFMLr showed a significant pH decrease by day two that remained roughly constant until the last day of fermentation (Figure 1B,D). On the other hand, in treatments without molasses (AwF and AwFLr, Figure 1A,C, respectively), the pH behavior was slightly different after day 8, showing a gradual increase until the last day of fermentation, although treatment AwFLr underwent a milder pH increase (Figure 1A,C, respectively). The addition of a starter culture had a significant (*p* < 0.05) influence on the treatments, which revealed drastic changes in pH drop throughout the fermentation (Figure 1B,D, respectively).

Simultaneously, bacterial counts were monitored for LAB and total coliforms. Treatments AwF and AwFM (Figure 1A,B, respectively), which did not have a bacterial starter, observed a significant (*p* < 0.05) increase in counts by day 2. Treatment AwF of LAB was 6.21 ± 0.163 logs CFU/g, while treatment AwFM was 5.6 ± 0.685 of CFU/g and remained approximately constant until the last day of fermentation for AwF treatment (6.03 ± 0.127 logs CFU/g) and a slight increase for the AwFM treatment (6.81 ± 0.041 logs CFU/g). For treatment AwFMLr (Figure 1D), a significant difference (*p* < 0.05) was observed between day 0 and day 8, but no difference was observed by day 14. This result contrasts with treatment AwFLr (Figure 1C), which had no significant difference in LAB counts, although some fluctuation was observed toward the end of the fermentation. Furthermore, the total coliforms for treatment AwF and AwFM also had similar trends, and a significant difference (*p* < 0.05) was found by day 2 with a sudden increase of counts, but afterward, only small changes were observed (Figure 1A,B, respectively). Treatment AwFLr (Figure 1C) presented different results with a significant increase of total coliforms by day 2 but also a drop by day 8 and a drastic and significant (*p* < 0.05) drop by day 14. Finally, treatment AwFMLr (Figure 1D) also showed a significant difference (*p* < 0.05) over time with essential changes of total coliforms. It is important to note that this last treatment presented zero counts of total coliforms after 8 days of fermentation.

### 3.2. Protein Dynamics Through Fermentation

Figure 2 shows the soluble protein concentration and the products of protein hydrolysis changes. The soluble protein indicated a significant (*p* < 0.05) decrease among all treatments from day 0 gradually toward the last day of fermentation. Moreover, the treatments containing the starter were more efficient in converting complex proteins into protein hydrolysates. Meanwhile, the concentration of the peptides increased significantly (*p* < 0.05) with a change of 5.24 ± 0.51 times more at day 14 than at day 0 for treatment AwF, 4.07 ± 0.61 for AwFM, 6.78 ± 0.24 for AwFLr, and 10.66 ± 0.42 for treatment AwFMLr showing consistency with the results of soluble protein hydrolysis and the efficiency of treatment AwFMLr.

The specific activity confirms these results since it showed a higher, but not significant (*p* < 0.05), activity within the treatments containing molasses and the LAB starter. Nevertheless, the treatment with the LAB starter (AwFLr and AwFMLr) had a significant (*p* < 0.05) influence over the yield of peptide concentration at the end of the fermentation as well as a more controlled and defined process. In comparison, the treatments that did not have a starter (AwF and AwFM) showed more variation in the yield of protein hydrolysates explained partially by the metabolism of the natural microbiota of the raw ingredients (Figure 3).

### 3.3. Metagenomic Analysis

A total of 16 samples were subjected to DNA extraction and 16S rRNA sequencing. Overall, 410, 694 (25,903.5 ± 6412.52) reads were obtained. All samples were rarefied to 8807 reads, and 69.05% of reads were mapped to the SILVA (v132) database Quast [28] and were used for further analyses.

Alpha diversity metrics were used to assess the microbial diversity within samples. The species diversity from the observed OUTs contained notably higher diversity in the control samples (AwF and AwFM) at day 0 than the samples from the remaining days (2, 8, 14) and the samples from the treatments that contained the starter (AwFLr and AwFMLr). Moreover, a decreased diversity was observed by day 2 in all treatments and followed a similar trend toward the last day; in addition, there was a slight difference among overall treatments that contain molasses (AwFM and AwFMLr) where less diversity was seen after day 2. The Shannon diversity index, which considers not only the species richness but also the evenness of the bacterial community, showed slightly different results than the ones described above. The treatments that had no starter (AwF and AwFM) increased the diversity by day 8, including the groups with and without molasses (AwF and AwFLr), and decreased by day 14. Unlike the treatments with the starter (AwFLr and AwFMLr) that presented lower diversity by day 8 than on day 2, the treatment that also had molasses (AwFMLr) showed a small diversity increase by the last day of the fermentation.

Furthermore, the beta diversity analysis carried out showed a notable separation of control samples (AwF and AwFM) when comparing the diversity, presence, and abundance of operational taxonomic units (OTUs) among the treatments. The phylogenetic distribution of the fermentation systems at the phylum level was first (day 0) lead by Firmicutes and Proteobacteria, which alone represented around 50–60% of the relative abundance of the bacterial communities. However, as the fermentation continued, Firmicutes outcompeted the remaining phyla until it dominated the system by day 14. It is important to mention that, as seen in Figure 4, molasses affects the relative abundance of bacterial diversity, as does the use of a starter. Other phyla seen across the samples are Fusobacteria and Bacteroidetes, especially in the treatments with no starter (AwF and AwFM).

Deeper in the taxonomy characterization, at the genus level, we focused on the top ten; however, Lactobacillus and Lactococcus were the most abundant, followed by *Pediococcus*, *Streptococcus*, *Cetobacterium*, *Rhodobacter*, *Pseudomonas*, *Chryseobacterium*, and *Enterococcus* (Figure 5). As expected by day 14, *Lactobacillus* is the predominant genus overall. However, as described above, the samples that did not contain the starter (AwF and AwFM) or the molasses had a more diverse genera distribution overtime. Only the treatment that contained molasses and starter behaved as expected (AwFMLr). Although the environmental conditions of development for these communities changed over time, it was also expected to be led by a lactose fermentative bacterium. Nonetheless, the species within these genera are well known to be heterofermentative; thus, a homogeneous community is desired.

## 4. Discussion

The pH of the acid whey and fish waste reported in this study were consistent with the work of Chandrapala [1] and Yang [30], respectively. The initial pH of the combined waste resulted in a slightly acidic pH (5.8). This pH drop is due to the effect of the lactic acid found in acid whey, as reported by Chandrapala [31]. Other authors have reported similar pH changes when using similar ingredients [32,33,34,35,36,37]. The initial pH allowed the development or inhibition of selected microorganisms. It has been reported that some LAB genera can grow at acidic pH [23,38,39]. On the other hand, some strains, such as *E. coli*, cannot grow at such acidic conditions, permitting the use of pH as a safety hurdle [33,34,40]. At the beginning (1–2 days) of the fermentation, there is a considerable pH drop that has also been reported in other studies [30,40,41,42]. This change is explained by the growth of certain bacteria species, particularly LAB, the growth of which was promoted in this study. LAB is known to produce lactic acid, among other bacterial metabolites, as products of fermentation [43]. The variability of time length to produce lactic acid depends on several factors, such as the availability of carbohydrates, temperature, overall composition, and microbial diversity, among other variables [33,37,40,44,45].

The composition of fish waste is rich in proteins but limited in carbohydrates [13,46]. Therefore, experiments have been conducted using molasses and other simple carbohydrates as sources because of their economic and environmental feasibility to achieve the products desired from similar fermentation processes [33,47]. In the making of fish silage, pH is an indicator not only of quality but also safety [35]. Furthermore, a pH lower than 4.5 is desired to have control over the pathogenic and spoilage microorganisms and favor the dominance of desired bacteria [16]. In this study, a highly proteolytic bacterium was chosen to accomplish two tasks: lower pH and hydrolyze complex proteins. *Lactobacillus rhamnosus* efficiently dropped the pH to the desired values and maintained it through the fermentation in one efficient treatment (AwFMLr). The treatments that did not have the starter or the carbohydrate source behaved differently. During the first half of the fermentation period, the pH decreased but did not reach the value that would be considered safe, which suggests that the lactose from the acid whey was used as a primary energy source during this first week. As a result, the production of lactic acid affected the pH and the production of protein hydrolysates with antimicrobial properties [23,46,47,48].

Interestingly, the treatment AwF that was used as the control and had no molasses or starter also showed a pH drop by the second day and maintained it during the first half of the fermentation. Then, the pH started to rise until it was slightly close to the initial pH of the mix. In this particular treatment, the endogenous microbial load from both ingredients was responsible for the pH drop and facilitated by the available lactose in the mix, followed by the increase of pH due to the release of compounds from the hydrolysis of proteins and peptides once the simple sugars were depleted [16,20,35]. Similarly, treatment AwFLr had two phases of pH behavior. However, the starter allowed it to reach a lower pH value in the first phase. The increase was higher than the initial pH of the mix, implying that a more significant proteolytic activity had occurred, and lactose was converted more efficiently to lactic acid.

The absence of total coliforms was seen only for treatment AwFMLr after the first week suggesting that the system was able to control pathogens, thus assuring its safety. The incorporation of fermentable sugars, as suggested by Javeed and Mahendrakar [48], increased the occurrence of organic acids, which inhibited the growth of pathogenic microorganisms. As reported by Kameník [49], some microorganisms are not tolerant of acidic conditions; therefore, in this study, pH was a parameter to manage its safety.

The use of a starter was noticeable when comparing the yields of products among treatments. *L. rhamnosus* is a Gram-positive, heterofermentative, facultatively anaerobic, non-spore-forming rod found as part of the gut microbiome. It is used in agriculture, dairy, and pharmaceutical industries due to its health benefits as well as its functionalities in fermentation [50,51,52]. This study supports the significance of the starter to obtain a semi-controlled protein hydrolysate production. Not only does it have an outstanding proteolytic activity in acidic environments, but it also exerts the production of lactic acid, which affects the pH modifying the environmental conditions of the fermentation and promoting the release of endogenous enzymes that facilitate the breakdown of complex proteins [46]—both sources of enzymes have been shown to work complementarily. The role of endogenous proteases to break down sarcoplasmic and myofibrillar proteins in an acidic pH as well as LAB proteolysis on these secondary peptides have been described in the literature [30]. Blending these to completely distant waste streams significantly surpasses each stream’s natural degradation independently. A recent publication that follows the natural degradation of fish waste from a necrobiome perspective describes natural fish decomposition as a source of putative pathogenic and toxigenic bacteria [53]. This work confirms their practical and complementary application for the potential valorization of waste from two food industries.

As expected, the metagenomic analyses indicated a higher diversity of the microbial community at the beginning of the fermentation. The treatment AwF was used as a control and showed a diverse relative abundance that was dominated by Firmicutes, Fusobacteria, and Proteobacteria. These phyla are known to be part of the microbiome of fish and water environments [54,55,56]. There was a significant difference in the overall diversity when adding the carbohydrate source as well as the starter. For example, as days passed in the fermentation, there was a clear dominance of the system by Firmicutes, which was anticipated because it is the phylum to which *L. rhamnosus* belongs [57]. These results are supported by nutrient availability and the changes in pH, which facilitate Firmicutes dominance. The involvement of fermentable sugars was noted to affect the relative abundance and diversity of the phylum because it modified the ecosystem that enabled favorable conditions for specific bacterial communities, as occurs in fermented foods [16,58,59,60]. It is important to note that the scarcity of simple carbohydrates leads bacteria to use proteins and peptides as alternative energy sources. Under such circumstances, the result is the production of nitrogenated compounds, increasing the pH and contributing favorable conditions for different bacterial communities, such as Bacteroidetes, among which several strains have been reported to exert proteolytic activity [30]. The treatment AwFM, regardless of the lack of a starter, modified its environment to favor the Firmicutes overgrowth that finally dominated the system. The environmental changes gave an advantage to the endogenous microbiota harboring the raw materials. However, the drop in pH was significant, allowing the growth of specific genera of Firmicutes. It was not an efficient microbial group for protein hydrolysis, and, therefore, a low yield of protein hydrolysates was achieved.

At the genus level among the top ten selected, *Lactobacillus* and *Lactococcus* dominated up to 90% of the community when starter and carbohydrate sources were given. Similar microbial shifts have been observed in fermented foods, such as sauerkraut, fish sauce, unpasteurized dairy products, and meats [16,35,38,51,58,60]. While the reported data are promising, further investigation is required to understand better the changes occurring during protein hydrolysis as well as nutrient bioavailability and its effects on the gut microbiome.

## 5. Conclusions

In this work, we present a viable process to increase the value of two waste streams of the food industry—acid whey from the processing of cheese and fish waste from fisheries’ muscle isolation. Partially degraded peptides resulted from the fermentation of these waste by-products with the supplementation of molasses and bacteria. The process demonstrated that with the minor addition of fermentable sugars and minimal temperature control, the proteins from both streams produce high yields of smaller molecular weight peptides.

Protein hydrolysis occurred independently of the use of sugars in the fermentation. The overall peptide yield was more efficient when sugar was present, resulting in a 20% higher product yield during protein digestion than the treatment containing only acid whey and fish. Similar results were presented using the LAB starter; however, a 5% lower yield of peptides was produced than the optimum treatment with both the starter and molasses. We also report coliform growth inhibition due to the rapid acidification bust by the addition of fermentable sugars.

The metagenomic analysis shows an interesting trend. Whether or not a LAB starter was used, the resulting domination of Firmicutes was similar in all treatments except for the ‘wild’ fermentation without sugar or starter. Among Firmicutes, Lactobacillus and Lactococcus genera represent more than 95% of the community. The presence of sugar and starter led to a faster normalization of microbial dominance. In the fermentation without sugar or starter, while Firmicutes were dominant, Proteobacteria and Bacteroidetes also had a significant presence. The conditions under which the fermentations were carried out had an impact on the microbial population. This specific microbiome is tailored to enhance the nutritious and safe product.

## Figures and Tables

**Figure 1 microorganisms-09-00100-f001:**
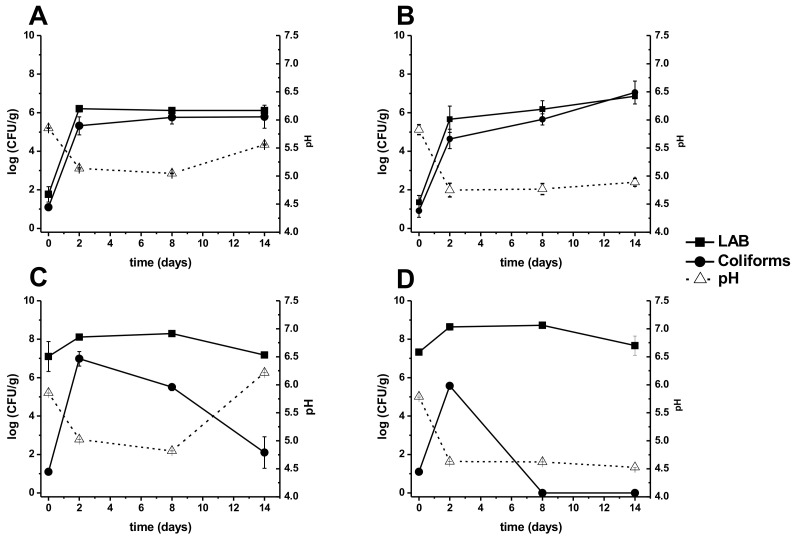
Microbiological and pH analyses: (**A**) acid whey fish (AwF); (**B**) acid whey fish molasses (AwFM); (**C**) acid whey fish lactic acid bacteria Lr (AwFLr); (**D**) acid whey fish molasses lactic acid bacteria Lr (AwFMLr). Filled squares represent lactic acid bacteria (LAB); filled circle represents coliforms; open triangles represent the pH. Error bars represent the standard deviation of three independent experiments.

**Figure 2 microorganisms-09-00100-f002:**
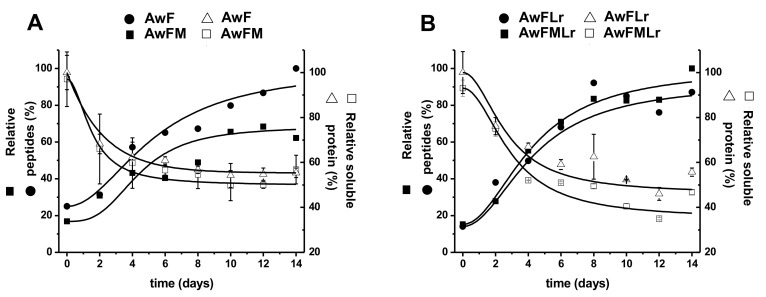
Soluble protein and peptide dynamics: (**A**) AwF and AwFM; (**B**) AwFLr and AwFMLr. Open squares represent relative soluble protein content; open triangles represent relative soluble protein content; filled squares represent relative peptide concentration; filled circles represent relative protein concentration. Error bars represent the standard deviation of three independent experiments.

**Figure 3 microorganisms-09-00100-f003:**
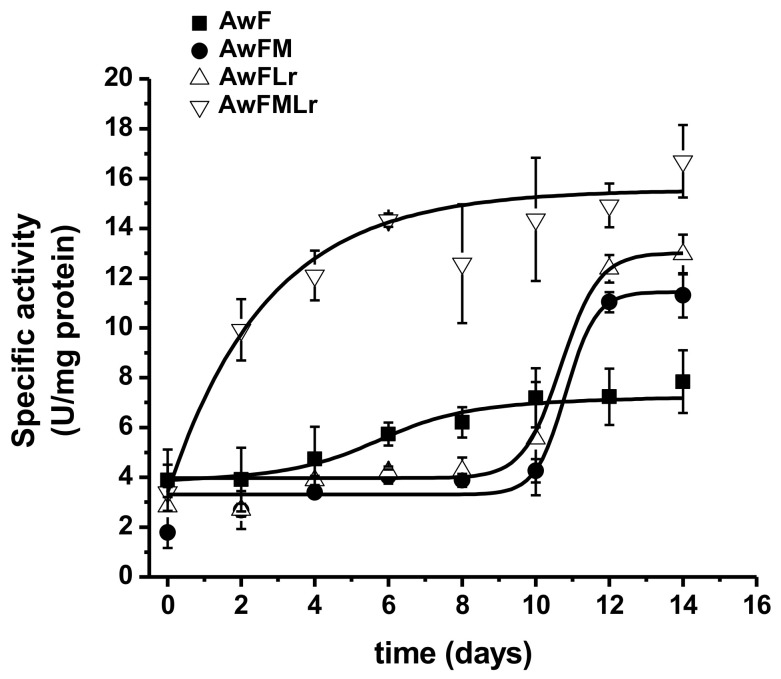
The specific activity of proteolytic dynamics.

**Figure 4 microorganisms-09-00100-f004:**
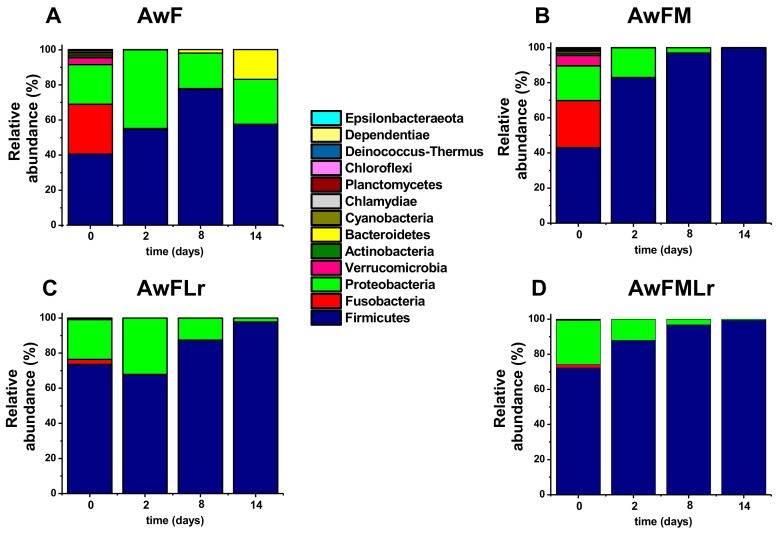
The relative abundance of the microbial community at the phylum level of the fermentation. (**A**) AwF; (**B**) AwFM; (**C**) AwFLr; (**D**) AwFMLr.

**Figure 5 microorganisms-09-00100-f005:**
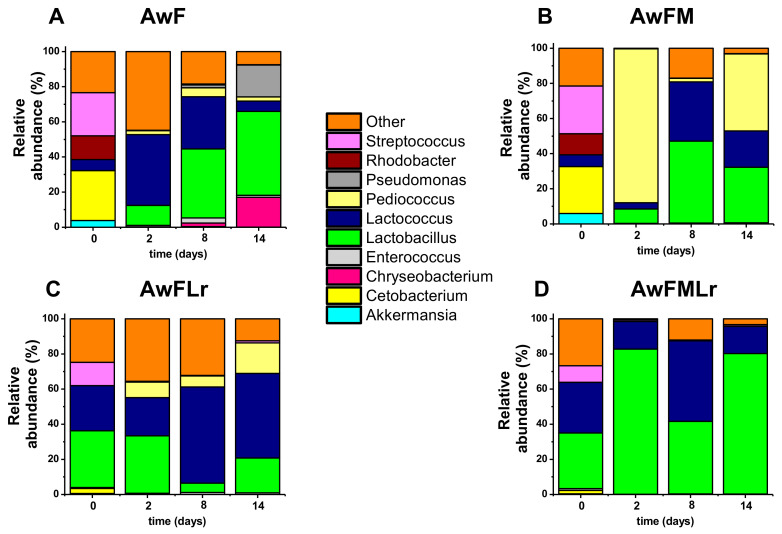
The relative abundance of the microbial community at the genus level of the fermentation. (**A**) AwF; (**B**) AwFM; (**C**) AwFLr; (**D**) AwFMLr.

**Table 1 microorganisms-09-00100-t001:** Experimental treatments.

			Fermentation Ratio			
Treatment	Content	Abbreviation	Acid Whey %	Fish %	Molasses %	LAB CFU/mL
1	Acid whey + Fish waste	AwF	50	50	-	-
2	Acid whey + Fish waste + Molasses	AwFM	47.5	50	2.5	-
3	Acid whey + Fish waste + *L. rhamnosus*	AwFLr	50	50	-	7.2 × 10^10^
4	Acid whey + Fish waste + Molasses + *L. rhamnosus*	AwFMLr	47.5	50	2.5	7.2 × 10^10^

## Data Availability

The 16S rRNA gene reads retrieved using Illumina pair-end sequencing form fermented product (raw data) have been deposited under the BioProject number PRJNA689224 in the NCBI Sequence Read Archive, with the accession numbers SAMN17203926- SAMN17203941.

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
