# Peer review of "Bacterial Diversity Analysis and Evaluation Proteins Hydrolysis during the Acid Whey and Fish Waste Fermentation"

_microorganisms, 2021, doi:10.3390/microorganisms9010100_

Round 1

Reviewer 1 Report

Main comments

  1. A proper description of the reactor is absolutely necessary:
  • What type of reactor was used?
  • What was the volume of the reactor?
  • Was it a batch culture?
  • Was it stirred?
  • Was it an anaerobic reactor?
  1. The description and motivation of the experimental design are missing. The experimental design and statistical analysis can be described together.
  2. The conclusions should provide a concise answer to the research question. For example, the effect of adding sugars is repeated in the conclusion (lines 384-385 and 396-397).

Minor comments

Line 2 (Title) Aanalysis should be Analysis

Line 89 were / was

Scientific names must be written with italics

Lines 187-188 Lactobacillus rhamnosus

Lines 281-283 Lactobaillus, Lactoccocus, others  

Lines 300 E. coli

Line 315 … rhamnosus

Line 338-339 Lactobacillus rhamnosus

Line 356 L. rhamnosus

Reviewer 2 Report

Although this work is well written, it is poor designed and the performed treatments are very limited and doesn’t reflect any novel information to the field. The conclusions is not sufficiently supported by well-designed experiments such as using various concentrations of the wastes and or molasses. The final results are very general and value and the importance of the final end products are needed to be focused.

The exact analysis of wastes, their components, CN ratio and all valid data for repeatability of such fermentations and impacts for other researchers are missing.

Besides some minor corrections in the text

L23 delete (respectively)

L188 strain name should be italic

L202-210 All data are not consistent with that described in the figures. The authors should confirm

L203 fig. 1c contain bacterial starter as shown in figure legend.

LAB count of log 8.0 or more should be reconfirmed from the author’s data

L300 E coli ; italic; L315,379. 356   L rhamnosus

Reviewer 3 Report

The manuscript submitted for review describes interesting and topical aspects related to the possibility of using various by-products of the food industry as a valuable source of nutrients with high potential applications. The aim of the research was to determine the microbiological diversity and to characterize the processes of protein decomposition during fermentation of a mixture of two by-products: acid whey and fish waste. The authors used Lb. rhamnosus with strong proteolytic properties demonstrated in previous studies.

I have no major comments on the chapters: Abstract, Introduction, Discussion and Conclusions. Minor comments to the text of these chapters are marked in the attached pdf file in the review mode.

I have some comments on the description of the methods used (lines 120 - 125) and the lack of description in the methodology of one type of results described in the Results chapter (total plate count - lines 202, 217-222).

Subsection 3.1. it should not be included in the Results chapter at all, but be part of the Materials and methods chapter.

One methodological objection may be the use of only MRS medium. The authors' assumption that all lactic bacteria present in the culture grow on this medium is unjustified. A detailed description in this regard is presented in the comment in line 122.

All other minor remarks have been included in the pdf file.

Round 2

Reviewer 3 Report

All comments and suggestions have been taken into account by the Authors.